# Use of *Ganoderma lucidum* (Ganodermataceae, Basidiomycota) as Radioprotector

**DOI:** 10.3390/nu12041143

**Published:** 2020-04-19

**Authors:** Aránzazu González, Violeta Atienza, Alegría Montoro, Jose M. Soriano

**Affiliations:** 1Food & Health Lab, Institute of Materials Science, University of Valencia, 46980 Paterna, Valencia, Spain; aranzazugonzalez999@gmail.com; 2Joint Research Unit on Endocrinology, Nutrition and Clinical Dietetics, University of Valencia-Health Research Institute La Fe, 46021 Valencia, Spain; 3Department of Botany and Geology, Faculty of Biological Sciences, University of Valencia, 46100 Burjassot, Valencia, Spain; M.Violeta.Atienza@uv.es; 4Radiation Protection Service, University and Polytechnic Hospital La Fe, 46021 Valencia, Spain; montoro_ale@gva.es; 5Biomedical Imaging Research Group GIBI230, Health Research Institute La Fe, 46021 Valencia, Spain

**Keywords:** *Ganoderma lucidum*, radioprotector, ex vivo, in vivo, in vitro, reishi mushroom, review

## Abstract

For millennia, naturopaths and physicians have used *Ganoderma lucidum* (reishi mushroom) for its diverse therapeutic properties, as recorded in the oldest Chinese herbal encyclopedia. Indeed, a radioprotective effect has been reported in the isolated components of its extracts. A systematic review and meta-analyses (PRISMA) was conducted in March 2020, searching databases including PubMed, Scopus, Embase, and Google Scholar, along with Clinical Trials. The inclusion criteria were ex vivo, in vitro, and in vivo studies, with full texts in English, conducted to determine the radioprotective benefits of *G. lucidum*, or reports in which ionizing radiation was used. From a total number of 1109 records identified, 15 full text articles were eligible, none of them were clinical trials. In vivo studies reveal the efficiency of *G. lucidum* aqueous extracts of polysaccharides and triterpenes in mice exposed to *γ*-rays. In plasmid, they can reduce radiation damage as an increment of the open circular form, as well as increase the DNA extension, as shown in vitro studies. Ex vivo studies conducted in human blood cells show the radioprotective effect of β-glucan of aqueous extract of *G. lucidum*, nevertheless, its implementation as radioprotector to humans is in need of further clinical research studies.

## 1. Introduction

*Ganoderma lucidum* (Curtis) P. Karst. (Ganodermataceae Polyporales) [1,2] is an annual wood-degrading Basidiomycota (Figure 1) that is frequently dimidiate, in the form of a stipulated shelf, hat or pileus of 40–200 (300) mm in diameter, irregularly rounded, oval, reniform or flabeliform (fan-shaped), and is frequently stipulated centrally or laterally. Table 1 shows some common names [3].

For habitat and distribution, *G. lucidum* is widely distributed in the world, typically in tropical and temperate regions in Europe, North America, and Asia. It lives as a saprotroph on a wide variety of trees from different families, mainly at the base of broad-leaved tree trunks, and more rarely on conifers. It is exceptionally parasitic, causing wood rot. In the natural environment it is a rare species that grows at the base of the trunks, even buried, of old trees, especially on maple [4]. *G. lucidum* has been recorded in the oldest Chinese herbal encyclopedia, called *Shen-Nong-Ben-Cao-Jing* (“The Herbal Classic of the Divine Plowman”) [5], edited during the Han Dynasty in China in about 2700 BC, and used in Oriental traditional medicine. Nowadays, it is utilized due to its beneficial effects, including antitumor [6], anti-angiogenic [7], anti-inflammatory [8], anti-herpetic [9], anti-microbial [10], anti-hypertensive [11], hypocholesterolemic [12], anti-histamine [13], anti-complement [14], hepatoprotective [15], and radioprotective [3] among other properties [16], this last effect being detected in the extracts and in isolated polysaccharide components [17]. The use of radioprotective compounds in research are applied in a radioactive environment, such as in space exploration, for individuals handling nuclear emergencies, radiation workers, and individuals subjected to diagnostic and therapeutic radiation exposures, as the compounds can minimize the effects of single-strand breaks, double-strand breaks (DSB), base damage and DNA–protein cross-links [18]. The aim of this systematic review was to evaluate the radioprotective effect of *G. lucidum*.

## 2. Methods

This systematic review was developed according to the Preferred Reporting Items for Systematic Reviews and Meta-Analyses (PRISMA) statement [19] (Figure 2) and conducted in March 2020 using PubMed, Scopus, Embase, and Google Scholar, along with www.clinicaltrials.gov, for completed or ongoing clinical trials. 

Authors decided not to limit the year of publication. Specifically, we used the keywords “radiation”, “radiation protection”, “radioprotector”, “*Ganoderma lucidum*”, “*Ganoderma*”, and common name in any language and any of their homotypic and heterotypic synonyms [3]; (*Agarico-igniarium trulla* Paulet., *Agaricus lignosus* Lam., *Agaricus pseudoboletus* Jacq., *Boletus castaneus* Weber., *Boletus crustatus* J.J. Planer., *Boletus dimidiatus* Thunb., *Boletus flabelliformis* Leyss., *Boletus laccatus* Timm., *Boletus lucidus* Curtis., *Boletus ramulosum* var. *flabelliformis* (Leyss.) J.F. Gmel., *Boletus rugosus* Jacq., *Boletus supinus* var. *castaneus* (Weber) J.F. Gmel., *Boletus verniceus* Brot., *Boletus vernicosus* Bergeret., *Fomes japonicus* (Fr.) Sacc., *Fomes lucidus* (Curtis) Sacc., *Ganoderma japonicum* (Fr.) Sawad., *Ganoderma mongolicum* Pilát., *Ganoderma nitens* Lázaro Ibiza., *Ganoderma ostreatum* Lázaro Ibiza., *Ganoderma pseudoboletus* (Jacq.) Murrill., *Grifola lucida* (Curtis) Gray., *Phaeoporus lucidus* (Curtis) J. Schröt., *Placodes lucidus* (Curtis) Quél., *Polyporus japonicus* Fr., *Polyporus laccatus* (Timm) Pers., *Polyporus laccatus* (Timm) Pers., *Polyporus lucidus* (Curtis) Fr., *Polyporus lucidus* var. *japonicus* (Fr.) Cleland & Cheel., and *Scindalma japonicum* (Fr.) Kuntze.). 

As inclusion criteria, we considered published ex vivo, in vitro, and in vivo studies [20], along with clinical trials with full texts that were conducted to determine the radioprotective effect of *G. lucidum,* and studies in English in which ionizing radiation was used. The following information was obtained: author name, year of publication, subject/cell/DNA, radiation type and dose, dose/concentration, and main outcomes. For clinical trials, if it was applicable, we used the Jadad scale [21]. On the other hand, for exclusion criteria the reasons were the following: 1. studies in which *G. lucidum* was not used; 2. *G. lucidum* used in combination with other agents; 3. studies that made use of other forms of radiation such as ultraviolet (UV), fluorescence, cosmic, etc.; 4. the effect of *G. lucidum* evaluated with chemotherapy instead of radiation therapy; 5. conference abstracts; 6. simulation studies; 7. review articles; 8. case reports; 9. letters; 10. editorials; 11. unpublished data; 12. articles without full texts; and 13. non-English articles. 

Two teams of paired reviewers (A.G., V.A., A.M., and J.M.S.) with expertise in medical and health evaluations, and training in research methodology, independently screened titles, abstracts, and full texts for eligibility, assessed generalizability, and collected data from each eligible study using standardized pilot tested forms with detailed instructions. Reviewers resolved disagreement through discussion.

## 3. Results

A total number of 1109 records were identified through database searching. After the screening of duplicates and irrelevant studies, 15 full text articles that were assessed remained eligible, all of which included in ex vivo, in vitro, and in vivo studies. None of the ten clinical trials, as reflected in www.clinicaltrials.gov, assayed *G. lucidum* as radioprotectors (Figure 2). Table 2, Table 3 and Table 4 demonstrate, respectively, ex vivo (3), in vivo (13), and in vitro (4) studies. 

The lowest number of studies was obtained using the ex vivo procedure due to technical difficulties in preparation and maintenance of isolated cells in comparison with other methods (in vivo and in vitro). Pillai et al. [22] used an aqueous extract of *G. lucidum* and observed a radioprotective effect, but it is important to relate that the major constituents of the aqueous extract of this mushroom are polysaccharides, as in β-glucan [23].

For in vivo studies (Table 3), two out of the 13 that were published used X-ray, being the most common γ-ray. The three most used preparations of *G. lucidum* are polysaccharides, aqueous extracts, and triterpenes, and all the literature from these studies show that their efficiency is in a dose-dependent manner and they are not toxic at the radioprotective dose. Gao et al. [36] indicated that the polysaccharides are among the major source of the pharmacologically active constituents in *G. lucidum*. In fact, the major constituent of the aqueous extract of this mushroom is polysaccharides [25]. The effectiveness of this long chains of carbohydrate molecule is observed in the reduction of malondialdehyde (MNA) [27,31], micronuclei induction [25,31], and in the increase in both the reduced glutathione (GSH) pool [26,27] and superoxide dismutase (SOD) activity [31]. For the triterpenes of *G. lucidum*, the radioprotective effect is observed with the reduction of micronucleated polychromatic erythrocytes (MCNE) [20], apoptotic cells [33], and reactive oxygen species (ROS) [33]; meanwhile, in aqueous extract, the effect exhibited is a reduction of lipid peroxidation [22]. 

For in vitro studies, the use of plasmid pBR322 DNA is viewed as tool with interest to assess the radioprotective effects due to radiation induced damage, as an increase in open circular form or decrease in supercoiled form of plasmid DNA. The application in plasmid of aqueous extract [22] or total triterpenes [20] of *G. lucidum* can reduce open circular form or increase DNA extension, respectively.

## 4. Discussion

Nowadays, the high global demand for *G. lucidum* is associated with its various pharmacological and therapeutic properties [37,38], focusing on its chemical compounds [39]. Briefly, the bioactive substances in the mushroom can be found in several of its parts, such as the mycelium and fruit body [4]. The most important types of these compounds are alkaloids, enzymes, glycoproteins, minerals, nucleotides, polysaccharides, proteins, steroids, triterpenes, and unsaturated fatty acids [39,40,41,42,43,44]. However, literature has reflected that the classification of the hydrophilic and hydrophobic properties of these compounds is useful to understand their effect as radioprotective compounds. Polysaccharides are hydrophilic [40], hence abundant in decoctions, whereas triterpenes are not [33]. In fact, Dai et al. [34] suggested that *G. lucidum* spore oil, which although it can be useful as a radioprotector, has the drawback of poor water solubility, which remains a major obstacle for their further development and clinical application in human health care. For this reason, these authors developed a GLSO@P188/PEG400 nanosystem (NS) to improve the efficiency of the radioprotective treatment, especially when this system has functional food composites with hydrophobic defects, such as triterpenes, but not polysaccharides.

On the other hand, a major drawback of studies using mushroom extract is that its chemical composition is not characterized. Extracts can have different chemical compounds and have different concentrations even between batches. This is important, as knowing that synergies or antagonisms can exist between each of the compounds, could have repercussions on the efficacy of the extract when studying it. According to the Montoro et al. [45], the characterization of the extracts should be an essential factor in all the studies of extracts. 

## 5. Conclusions

Ex vivo studies conducted in human blood cells (leukocytes, and human peripheral blood lymphocytes) show the radioprotective effect of β-glucan of aqueous extract of *G. lucidum* against γ-ray radiation-induced damage. In plasmid, they can reduce radiation damage as an increment of the open circular form, as well as increase the DNA extension, as shown in vitro studies. In vivo studies reveal the efficiency of *G. lucidum* aqueous extracts of polysaccharides and triterpenes in mice exposed to *γ*-rays. Yet we must not forget that the doses of these compounds or the radiation used in the reviewed studies cannot be directly correlated to humans; therefore, further studies are required for its clinical implementation as a radioprotector.

## Figures and Tables

**Figure 1 nutrients-12-01143-f001:**
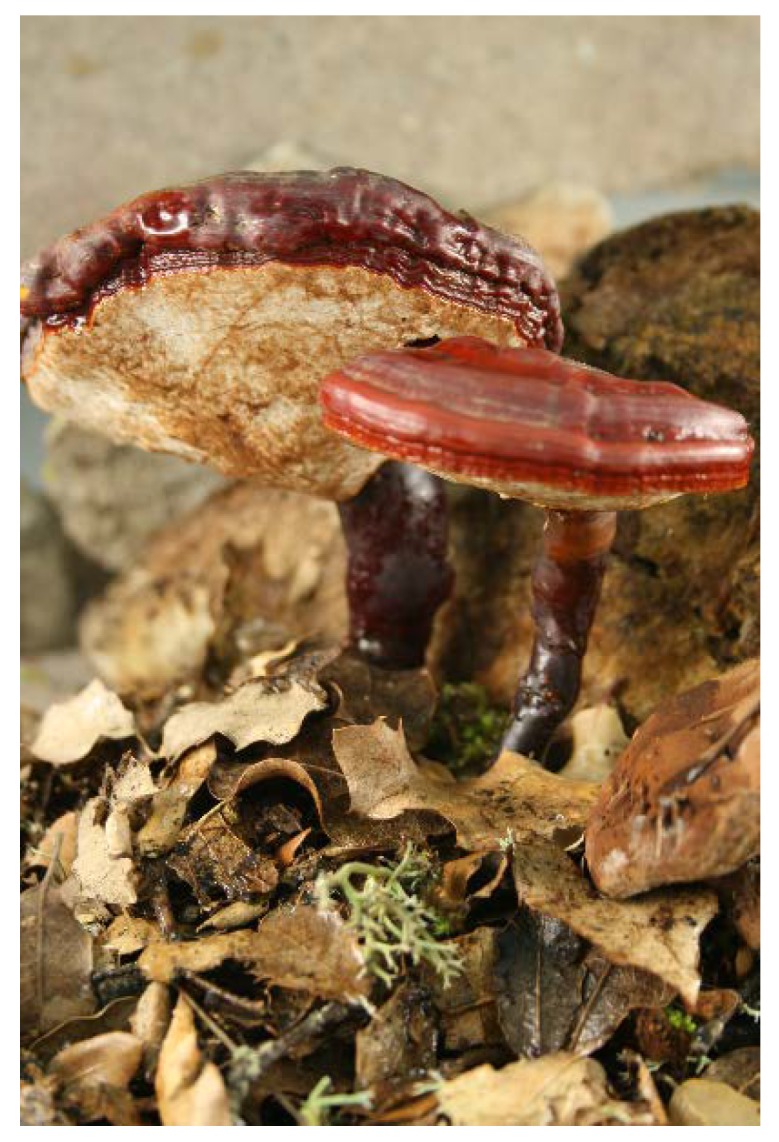
*Ganoderma lucidum* at the base of *Quercus coccifera* in Spain, Castellón, El Toro, 19 November, 2008, Mir and Atienza, Herbarium VAL-Myco 134.

**Figure 2 nutrients-12-01143-f002:**
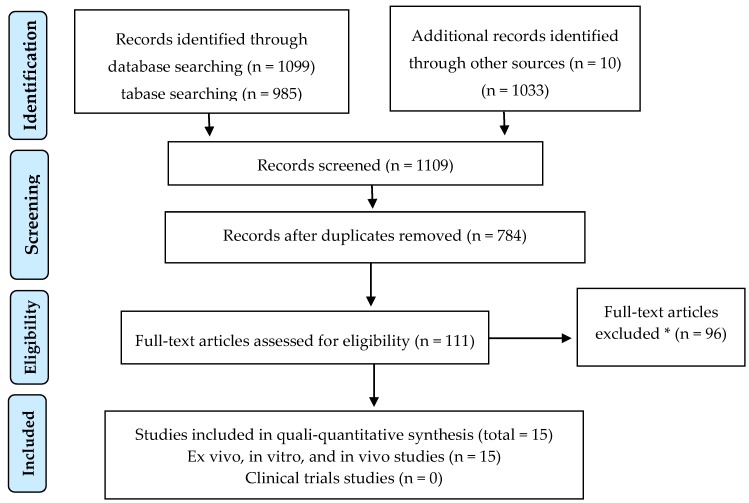
PRISMA (Preferred Reporting Items for Systematic Reviews and Meta-Analyses) flow diagram for studies retrieved through the searching and selection process. * The reasons for the exclusion of articles were the lack of critical information and methodological shortcomings (see Methods section).

**Table 1 nutrients-12-01143-t001:** Common name of *Ganoderma lucidum* in several languages, adapted from [3].

Language	Local Name	Other Characteristics
Spanish	Pipa	-
Catalan	Paella, pipa.	-
Chinese (Pinyin)	靈芝 *(Lingzhi)* is first collected during the Eastern Han Dynasty (AD 25–220). *Lingzhi* is composed of 靈 *(ling)* which means “spirit”, supernatural, soul, miracle, sacred, divine; and 芝 *(zhi)* is a word that it does not have an exact translation in non-eastern languages and refers to a set of varied objects such as plants, fungi, mushrooms, and seeds.	In China some *Ganoderma* species are differentiated as 赤芝 (chizhi), “Red mushroom”; or *G. lucidum* and 紫芝 (zizhi), “Purple mushroom”. Other Chinese names are 瑞芝 (ruizhi), “Propitious mushroom”; 神 芝 (shenzhi), “Divine mushroom”; 木 靈芝 (mulingzhi), refers to tree or wood; 仙草 (xiancao), “Plant of immortality or immortal”; and 靈芝 草 (lingzhicao) or 芝 草 (zhicao), “Mushroom plant”.
English	Glossy ganoderma, shiny polyporus.	-
French	Polypore luisant, ganoderme luisant.	-
German	Lackporling, glänzender lackporling.	-
Japanese	Reishi (霊 芝).	Other literary terms for this mushroom are zuisō (瑞草), “Propitious plant”; and sensō (仙草), “Plant of immortality”. A name used is mannentake (万年 茸), “10,000-year-old mushroom”. Written Japanese uses shi or shiba (芝) for grass and uses take o kinoko (茸) for “mushroom.”
Korean	Yeong Ji or Yung Gee (영지, 靈芝). It is also called Seon-cho (선초, 仙草), Gil-sang-beo-seot (길상 버섯, 吉祥 茸), Yeong ji cho (영지 초, 靈芝 草) or Jeok hee (적지, 赤芝).	It is named according to their colors: Ja-ji (자지, 紫芝), if it is purple; Heuk-ji (흑지, 黑 芝), black; Cheong-ji (청지, 靑 芝), blue or green; Baek-ji (백지, 白 芝), white; Hwang-ji (황지, 黃 芝), yellow.
Vietnamese	Linh chi	Often used with (nấm Linh Chi), which is the equivalent of *G. lucidum* or reishi mushroom

**Table 2 nutrients-12-01143-t002:** Summary of articles focused on the ex vivo radioprotective effect of *G. lucidum* included in the systematic review.

Cell	Radiation Type and Dose (Gy)	Dose/Concentration	Main Outcomes	Reference
Leukocytes	γ-ray, 50	50 μg/mL aqueous extract of *G. lucidum*	β-glucan protected DNA against radiation-induced single-strand breaks; reduced the increase in % tail DNA, tail length, tail moment and olive tail moment.	[22]
Human peripheral blood lymphocytes	γ-ray, 0, 1, 2, and 4	50 and 100 μg/mL β-glucan of *G. lucidum*	Reduction of comet parameters, such as the % tail DNA, tail length, tail moment and olive tail moment.	[23]
Human peripheral blood lymphocytes	γ-ray, 2	10, 50, and 100 μg/mL aqueous extract of *G. lucidum*	Reduction of the % tail DNA, tail length, tail moment and olive tail moment.	[20]

**Table 3 nutrients-12-01143-t003:** Summary of articles focused on the in vivo radioprotective effect of *G. lucidum* included in the systematic review.

Subject (Weight)	Radiation Type and Dose (Gy)	Dose/Concentration	Main Outcomes	Reference
ICR old male mice, 6 to 7 weeks old	X-ray, 500 or 650 cGy	Hydro-alcoholic extract of *G. lucidum*	Recovered the body weights and increased the recovery of hemograms of radio-irradiation. However, the differences of the radioprotective effect between the X-ray irradiated groups with *G. lucidum* pre- and post-treated were not significant.	[24]
Male Swiss albino mice, 8–10 weeks old (20–25 g)	γ-ray, at 350 Gy	50 μg/mL aqueous extract of *G. lucidum*	β-glucan of *G. lucidum* prevented 98% of lipid peroxidation.	[22]
Male Swiss albino mice, 8–10 weeks old (20–25 g)	γ-ray, 10	250 and 500 µg/kg body weight of aqueous extract of *G. lucidum*	At a dose of 500 µg/kg body weight, the polysaccharides were most effective in protecting animals from radiation induced loss of lethality. Furthermore, the decrease in micronuclei induction was dose dependent.	[25]
Swiss albino mice, 6–8 weeks old (24–28 g)	γ-ray, 4	10 and 20 mg/kg body weight of aqueous extract of *G. lucidum*	The depleted level of GSH in the jejuna mucosa was restored significantly by the aqueous extract of *G. lucidum*.	[26]
Swiss albino mice, 6–8 weeks old (26–30 g)	γ-ray, 4	10 and 20 mg/kg body weight of aqueous extract of *G. lucidum*	Reduction of the serum MDA levels compared to the irradiated group. Tissue GSH was maintained at normal levels after administration of polysaccharides.	[27]
Swiss albino mice, 6–8 weeks old (28–32 g)	γ-ray, 4 and 8	250 and 500 µg/kg body weight of β-glucan of *G. lucidum*	Significant reduction in the number of aberrant cells and different types of aberration, including polyploidy and cells with pulverization, were observed in both β-glucan administration.	[28]
Swiss albino mice, 8–10 weeks old (22–25 g)	γ-ray, 4	200 µg/kg body weight of aqueous extract of *G. lucidum*	Protection to normal tissues against gamma radiation-induced DNA damage, whereas in sparing tumor tissues, the extract offered no protection against radiation-induced cellular DNA damage.	[29]
Male Swiss albino mice, 6–8 weeks old (22–25 g)	γ-ray, 8	100 mg/kg body weight of hydro-alcoholic extract of *G. lucidum*	Considerable protection of DNA in blood leucocytes, bone marrow cells, brain cells, and intestine cells. The protection of brain tissue DNA from radiation-induced damage indicates that extract, or its biological active components, do not cross blood brain barrier. Extract administration bestowed survival advantage for mice following whole-body lethal ionizing radiation exposure.	[30]
ICR female mice, 6–8 weeks old (18–22 g)	γ-ray, 3 and 6	13.4, 26.6, and 40.0 mg/kg of aqueous extract of *G. lucidum*	Aqueous extract of *G. lucidum* did not reduce the impact of radiation on WBC levels, but 26.6 and 40 mg/kg of aqueous extract of *G. lucidum* treatment demonstrated that MDA levels were significantly decreased, the SOD activity was restored to near normal levels, the micronuclei frequency was reduced, and the nucleated cell count in bone marrow was significantly increased by aqueous extract of *G. lucidum* treatment in a dose-dependent manner.	[31]
Adult female Swiss albino mice (22–25 g) with tumor inoculation *	γ-ray, 3 × 2 Gy at two days interval to attain a total dose of 6 Gy	100 mg/kg body weight of FGL	Elevation in the concentration of MDA accompanied by a decrease in SOD activity and GSH content in liver tissues. A remarkable increase was observed in AFP and IL-2 concentration in serum.	[32]
Male Swiss albino mice (23–27)	γ-ray, 2.5	50 and 100 mg/kg body weight of total triterpenes of *G. lucidum*	The treatment with 100 mg/kg body weight of total triterpenes effectively reduced the percentage of MNPCE nearly to normal levels.	[20]
Male Swiss albino mice (23–27 g)	γ-ray, 2	25, 50, and 100 µg total triterpenes of *G. lucidum*	Effective in preventing DNA laddering and DNA damage; reduced apoptotic cells and the formation of intracellular ROS. Furthermore, endogenous antioxidant enzyme activity was enhanced in the splenic lymphocytes following irradiation.	[33]
BALB/c nude mice, 5 to 6 weeks old (18–22 g)	X-ray, 16 or 20 Gy	GLSO@P188/PEG_400_ nanosystem (NS)	This NS could reverse X-ray-induced cardio dysfunction, improve long-term renovation processes, and attenuate chronic cardiac fibrosis and necrosis from X-rays.	[34]

***** The diameter of the tumor reached approximately (10 mm); AFP: alpha-fetoprotein, FGL: fermentation filtrate of *G. lucidum,* GSH: reduced glutathione, IL-2: interleukin-2, MNPCE: micronucleated polychromatic erythrocytes, MDA: malondialdehyde, ROS: reactive oxygen species, and SOD: superoxide dismutase.

**Table 4 nutrients-12-01143-t004:** Summary of articles focused in vitro radioprotective effect of *Ganoderma lucidum* included in the systematic review.

Cell/DNA	Radiation Type and Dose (Gy)	Dose/Concentration	Main Outcomes	Reference
Plasmid pBR322 DNA	γ-ray, 25	1, 5, 10, 25 and 50 μg total triterpenes of *G. lucidum*	Reduction in the open circular form in a dose-dependent manner which obtained a retention of 98.87% of the supercoiled form with 50 μg total triterpenes.	[20]
Plasmid pBR322 DNA	γ-ray, 50	50 μg/mL aqueous extract of *G. lucidum*	Protection to the plasmid DNA to an extent of 89.53%.	[22]
Rat cardiomyocytes (H9C2 cells)	X-ray, 2, 8, 16 Gy	GLSO@P188/PEG_400_ nanosystem (NS)	Any post-treated strategy after X-ray irradiation (repair strategy) exhibited relatively inefficient effects, compared with pre-treated strategies on H9C2 cells from X-rays. The ideal strategy of pre-treated GLSO@P188/PEG_400_ NS before irradiation for 4–8 h showed an efficient protection effect on H9C2 cells from X-rays (16 Gy), leading to an increase of cell viability of 101.4%–112.3%	[34]
M13mpl9 RF DNA	X-ray, 10, 20 and 30 Gy	Hot-water extract of *G. lucidum*	Protection against hydroxyl radical-induced DNA strand breaks	[35]

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
