# Peer review of "Use of *Ganoderma lucidum* (Ganodermataceae, Basidiomycota) as Radioprotector"

_nutrients, 2020, doi:10.3390/nu12041143_

Round 1
Reviewer 1 Report
Lin 33 to 46: this part of introduction contains primitive data about the morphological properties of the G. lucidum. I suggest to rewrite this introduction part according to the aims of the study.
Line 52: “several”; but there are only some names
Line 52: table 1. I suggest to write ( local name) or (common names) instead “term”.
Line 60: “Oriental medicine”; did you mean (Oriental traditional medicine)?
Line 60-63: “including antitumor [6], anti-angiogenic [7], anti-inflammatory [8], anti-herpetic [9], anti-microbial [10], anti-hypertensive [11], hypocholesterolemic [12], anti-histamine [13], anti-complement [14], hepatoprotective [15] and radioprotective [3]”. But all these terms are new pharmacological terms how thy including in old traditional medicine?
Line 120: why table 3 is separated in some tables. Same for Table 2.
Line 130: table title: “Summary of articles focused ex vivo radioprotective effect of G. lucidum”. Which part of basidiomycete is extracted? Which extract solvent is used, or which extract type used.
Line 132: “10, 50 and 100 μg total triterpenes of G. lucidum”. This is not clear if it is extract or compounds. Check across tables.
Line 134: Table 3. “GLC”. Which type of extract? Which dose?
Line 134: Table 3 “GLP”, what is used: pure compounds (mixture or single) or extract. Check across tables
Author Response
Reviewer’s comment: Lin 33 to 46: this part of introduction contains primitive data about the morphological properties of the G. lucidum. I suggest to rewrite this introduction part according to the aims of the study.
Author’s comment: We have decided to delete this part but we think is important to describe briefly the form of G. lucidum and we have left from line 33 to line 36.
Reviewer’s comment: Line 52: “several”; but there are only some names
Author’s comment: we have changed this part by “Table 1 shows some common names” to clarify it.
Reviewer’s comment: Line 52: table 1. I suggest to write ( local name) or (common names) instead “term”.
Author’s comment: we have changed by “local name”.
Reviewer’s comment: Line 60: “Oriental medicine”; did you mean (Oriental traditional medicine)?
Author’s comment: we have changed by “Oriental traditional medicine”.
Reviewer’s comment: Line 60-63: “including antitumor [6], anti-angiogenic [7], anti-inflammatory [8], anti-herpetic [9], anti-microbial [10], anti-hypertensive [11], hypocholesterolemic [12], anti-histamine [13], anti-complement [14], hepatoprotective [15] and radioprotective [3]”. But all these terms are new pharmacological terms how thy including in old traditional medicine?
Author’s comment: This part has been changed in order to avoid confusion.
Reviewer’s comment: Line 120: why table 3 is separated in some tables. Same for Table 2.
Author’s comment: We have thing to separate due to that are different tables. Tables 2, 3 and 4 are summary of articles focused ex vivo, in vivo and in vitro, respectively, radioprotective effect of Ganoderma lucidum included in the systematic review.
Reviewer’s comment: Line 130: table title: “Summary of articles focused ex vivo radioprotective effect of G. lucidum”. Which part of basidiomycete is extracted? Which extract solvent is used, or which extract type used.
Author’s comment: Part of basidiomycete used in the references 20, 22 and 23 not be indicated in the articles. As example, reference 22 articles literally said that “The extract of G. lucidum was obtained from mushrooms collected from the outskirts of Thrissur, Kerala, India. The type specimen was deposited in the herbarium of Centre for Advanced Studies in Botany, University of Madras, Chennai, India (HERB. MUBL.3175).”
On the other hand, the extract type used is not indicated, as example in the reference 22 reflected only that is aqueous extract.
Reviewer’s comment: Line 132: “10, 50 and 100 μg/ml total triterpenes of G. lucidum”. This is not clear if it is extract or compounds. Check across tables.
Author’s comment: It is extract. It has been changed in the Line 132.
Reviewer’s comment: Line 134: Table 3. “GLC”. Which type of extract? Which dose?
Author’s comment: It is a hydro-alcoholic extract but not indicated the dose.
Reviewer’s comment: Line 134: Table 3 “GLP”, what is used: pure compounds (mixture or single) or extract. Check across tables
Author’s comment: It is a very interesting comment. It must be seen in the method section of the manuscripts being indicated the following procedure: “The powder was dissolved in double distilled water and administered orally in the experiments”. For this reason, we have changed GLP by “aqueous extract of G. lucidum”.

Reviewer 2 Report
Comments
Manuscript I.D. nutrients-769940
Title: Use of Ganoderma lucidum as radioprotector
Corresponding authors: Jose M. Soriano
Comments:
The review manuscript systematically collected articles studying the radioprotective effect of Gonoderma lucidum where ionizing radiation was used. Some criteria were setting to qualify the searching articles such as excluding of conference abstracts/case reports/letters/editorials/unpublished data/ non-English articles etc. From a total number of 1109 records identified, only 15 full articles were eligible and none of them were clinical trials. Following, the review discussed the effect of G. lucidum in ex. vivo, in vivo, and in vitro, respectively. From the rearrangements (Table 2-4), G. lucidum extracts, polysaccharides, or triterpenoids of G. lucidum do possess potent radioprotective effect. The review has scientific value and would attract some readers. Especially, relatively little studies focused on this field. Therefore, the reviewer supports its publication.
Author Response
Reviewer’s comment: The review manuscript systematically collected articles studying the radioprotective effect of Gonoderma lucidum where ionizing radiation was used. Some criteria were setting to qualify the searching articles such as excluding of conference abstracts/case reports/letters/editorials/unpublished data/ non-English articles etc. From a total number of 1109 records identified, only 15 full articles were eligible and none of them were clinical trials. Following, the review discussed the effect of G. lucidum in ex. vivo, in vivo, and in vitro, respectively. From the rearrangements (Table 2-4), G. lucidum extracts, polysaccharides, or triterpenoids of G. lucidum do possess potent radioprotective effect. The review has scientific value and would attract some readers. Especially, relatively little studies focused on this field. Therefore, the reviewer supports its publication.
Author’s comment: Authors thank the reviewer’s comment.
